# Graphene-Doped Thermoplastic Polyurethane Nanocomposite Film-Based Triboelectric Nanogenerator for Self-Powered Sport Sensor

**DOI:** 10.3390/nano14191549

**Published:** 2024-09-25

**Authors:** Shujie Yang, Tatiana Larionova, Ilya Kobykhno, Victor Klinkov, Svetlana Shalnova, Oleg Tolochko

**Affiliations:** 1Department of Physics and Materials Technology, Institute of Machinery, Materials and Transport, Peter the Great St. Petersburg Polytechnic University, Saint Petersburg 195251, Russia; yan6.sh@edu.spbstu.ru (S.Y.);; 2World-Class Research Center “Advanced Digital Technologies”, State Marine Technical University, Saint Petersburg 190121, Russia

**Keywords:** triboelectric nanogenerator (TENG), self-powered sensor, nanomaterials, graphene

## Abstract

Triboelectric nanogenerators (TENGs), as novel electronic devices for converting mechanical energy into electrical energy, are better suited as signal-testing sensors or as components within larger wearable Internet of Things (IoT) or Artificial Intelligence (AI) systems, where they handle small-device power supply and signal acquisition. Consequently, TENGs hold promising applications in self-powered sensor technology. As global energy supplies become increasingly tight, research into self-powered sensors has become critical. This study presents a self-powered sport sensor system utilizing a triboelectric nanogenerator (TENG), which incorporates a thermoplastic polyurethane (TPU) film doped with graphene and polytetrafluoroethylene (PTFE) as friction materials. The graphene-doped TPU nanocomposite film-based TENG (GT-TENG) demonstrates excellent working durability. Furthermore, the GT-TENG not only consistently powers an LED but also supplies energy to a sports timer and an electronic watch. It serves additionally as a self-powered sensor for monitoring human movement. The design of this self-powered motion sensor system effectively harnesses human kinetic energy, integrating it seamlessly with sport sensing capabilities.

## 1. Introduction

The advancement of the Internet of Things (IoT) has significantly increased the application of sensing devices in both industrial and everyday settings. Relying on traditional power supply methods for these devices often results in inefficiencies and high operational costs [1,2,3]. Consequently, the development of self-powered sensor devices has become increasingly imperative [4]. Harnessing abundant mechanical energy is regarded as one of the most effective strategies for powering self-powered electronic devices [5,6]. Triboelectric nanogenerators (TENGs), which utilize the coupling of triboelectrification and electrostatic induction effects, have emerged as a promising solution for sustainable energy generation [7].

TENGs harness various low-frequency mechanical energies from the environment, such as human kinetic energy, mechanical vibration energy, and wind energy [8,9,10,11]. TENGs have garnered extensive attention and undergone in-depth research worldwide due to their notable advantages, including a simple structure [12], low fabrication costs [13], and a diverse selection of materials [14,15]. Since the concept of TENGs was first introduced in 2012 [16], research has deepened, yielding significant breakthroughs in understanding the theoretical foundations of TENGs. Furthermore, research on the practical applications of TENGs has expanded, becoming more extensive and comprehensive.

Professor Zhonglin Wang’s comprehensive analysis of TENG theory has established that its fundamental principle is derived from Maxwell’s displacement current [17]. By applying Maxwell’s displacement current theory, various TENG operational modes have been effectively modeled. Thus, TENGs represent a practical application of Maxwell’s displacement current in both energy generation and sensor technology [18]. The underlying physical mechanism of the triboelectrification effect has been subject to extensive debate. Recent studies have identified spectra of atomic features during the triboelectrification process between two solid materials [19]. This atomic feature spectrum provides compelling evidence for electron transfer from atoms in one material to those in another at the contact electrification interface, firmly establishing that electron transfer predominantly governs the triboelectrification effect between solid materials [20].

As a leading technology in energy harvesting, TENGs are increasingly utilized in micro- and nano-distributed energy systems, self-driven sensing systems, blue energy, and high-voltage power supplies. Wei [21] developed an all-weather droplet-based TENG that captures electrical energy from solid–liquid contact interactions, serving as a power source for electronic devices. Wang [22] introduced a rotation-driven instantaneous discharging TENG (RDID-TENG), capable of driving a laser for environmental sensing and supporting wireless communication while delivering excellent electrical output. Pang [23] constructed a segmented swing-structured fur-based TENG (SSF-TENG) designed to harvest low-frequency water wave energy. Furthermore, Liu [24] pioneered the use of a single electrode mode TENG to generate ultra-high-voltage power, achieving a record output of 12.7 kV.

TENGs have shown significant progress in research related to micro–nano power sources and self-driven sensing [25,26,27]. The electrical output performance of TENGs is highly dependent on the selection of friction materials, and nanomaterial doping to prepare composite films is widely used in TENG design. The primary functions of doping nanomaterials into the friction layer of a TENG are as follows: (1) to enhance the triboelectric effect between the friction layer materials and improve the charge transfer capability of the layer; (2) to increase charge density by expanding the surface area of the friction layer, thereby raising the charge density per unit area; and (3) to improve the mechanical properties of the material, as the incorporation of nanomaterials can bolster the strength and wear resistance of the friction layer, thereby enhancing its stability and longevity during operation. Zu [28] prepared a wearable TENG by doping MoS2/GO nanoparticles into a silicone rubber matrix, revealing well device output performance. Chen [29] improved the anomalous output performance of a designed TENG by doping Sb into the friction layer. Huang [30] investigated composite friction layers doped with graphene, which exhibited remarkably high open-circuit voltage, short-circuit current, and power density.

While improving the output performance of TENGs by means of nanomaterial doping, the operational stability and durability of the composite film are also important factors to be considered. Friction layers with excellent durability can endure more friction cycles and wear, thereby extending the service life of the TENG. Enhanced wear resistance reduces the need for frequent replacements or repairs, saving both time and resources. By improving the durability of the friction layer, the overall performance and utility of the TENG can be significantly enhanced, ensuring its long-term stability and efficient operation in practical applications. The extensible TENG prepared by Zu [28] remained stable after 6000 cycles of test. Cao [7] designed a TENG based on crumpled MXene film, conducting excellent cycling stability.

However, there remain areas requiring further improvement. Currently, polydimethylsiloxane (PDMS) is frequently used due to its effective electrical output in TENG applications [31]. Nevertheless, the poor abrasion resistance and malleability of PDMS compromise the long-term stability and durability of TENGs that incorporate PDMS as the friction material. Additionally, the structural design of TENGs requires enhancement. Although the performance output of TENGs is often excellent, their structural complexity and the cost of materials pose challenges. Efforts to simplify the design and reduce material costs are essential to enhance the viability of TENGs for widespread application.

In this study, a graphene-doped thermoplastic polyurethane (TPU) nanocomposite film-based TENG (GT-TENG) was developed and evaluated for self-powered motion-sensing applications. TPU is recognized for its environmental friendliness and high performance as an elastomeric material. It exhibits a diverse range of hardness levels, outstanding mechanical properties, excellent transparency, and robust resistance to oil, water, and weathering. The GT-TENG features a straightforward design, low fabrication costs, sustained stability in electrical output, and exceptional durability. These characteristics render it highly suitable for applications in self-powered energy sensors, where it shows considerable potential.

This study presents the fabrication and evaluation of a GT-TENG designed for self-powered sport sensing. The article initially details the preparation process of the GT-TENG, followed by experimental validation. The results demonstrate that the GT-TENG can effectively convert mechanical energy into electrical energy, which is then used to power an LED, a sports timer, and an electric watch. Additionally, the GT-TENG functions as a self-driven sensor capable of monitoring human movement. Enhancements in the circuit design enable the GT-TENG to transmit the electrical signals via Bluetooth connection to computers and cell phones, facilitating wireless sport sensing monitoring.

## 2. Materials and Methods

### 2.1. Materials

In this study, thermoplastic polyurethane (TPU) granules Vitur TM1413-85, with a density of 1.177 g/cm^3^ and Tg = −50 °C, were used as row material for the polymer matrix for creating composites; the solvent was N, N-dimethylformamide (DMF) with the purity not less than 99.9%. As a filler, the graphene particles with a thickness of <100 nm and comparison area of 50–200 m^2^/g, sourced from Leader Nano Technology Company, Jinin, China, were used. The friction layer utilizes polytetrafluoroethylene (PTFE), measuring 2 cm × 2 cm. TPU-based composites and PTFE were paired with conductive aluminum tape of the same size on the back side, featuring a thickness of 70 µm to enhance conductivity. Both materials were mounted on a polymethylmethacrylate (PMMA) substrate to form the GT-TENG model, which served as the experimental platform for investigating triboelectric effects facilitated by these advanced material combinations.

### 2.2. Preparation of Graphene-Doped TPU Nanocomposite Films

The fabrication process of graphene-doped TPU composite films is outlined in Figure 1. Initially, TPU particles were dissolved in DMF solvent and left to stand for 36 h to ensure complete dissolution. Graphene nanoparticles, in varying proportions [0.5%, 1%, 1.5%, 2%], were then weighed and thoroughly mixed into the DMF. This graphene-infused DMF was subsequently combined with the DMF + TPU solution, followed by mechanical stirring. The mixture was then subjected to ultrasonication for 30 min to achieve uniform dispersion of graphene within the TPU solution. Approximately 3 mL of this homogenized solution was deposited onto a 12 cm × 12 cm quartz plate using a rubber-tipped burette. A glass rod was used to ensure even distribution of the solution across the plate. The coated quartz plate was then placed in a desiccator and heated to 150 °C for 5 min to completely volatilize the DMF, resulting in a single-layer TPU + Graphene film. This procedure was repeated to stack multiple layers of the TPU + Graphene films in a hot press mold, with a 250 µm metal shim inserted to maintain film thickness uniformity. The press was set to apply 4 kPa at 180 °C for 15 min. The pressed samples were allowed to cool until the mold temperature reached below 80 °C, producing graphene-doped TPU films with a thickness of 250 µm. The final film samples are depicted in Figure 1g. Figure 1i shows the microstructure of graphene nanoparticles under a scanning electron microscope (SEM-Mira3, Tescan, Brno, Czech Republic), while the SEM photograph of graphene nanoparticle distribution in the TPU matrix is shown in Figure 1h.

### 2.3. Assembly and Working Principle of TENG

Conductive aluminum tape, measuring 2 cm × 2 cm, was affixed to one side of both the PTFE and the graphene-doped TPU film. This aluminum layer served as the electrode for the GT-TENG, while the surfaces of the PTFE and TPU films without attached aluminum tape functioned as the friction layers. Each layer was connected to its respective electrode via a conductor. Two PMMA plates were utilized as the substrate, and four springs (diameter 5 mm, 15 mm height) were positioned at the corners of the plates to facilitate elastic contact and separation of the GT-TENG friction materials. Double-sided adhesive was placed between the electrodes and the substrate to maintain a 3 mm separation between the two friction layers in their initial state. A custom-designed mechanical reciprocating motion system was employed to subject the GT-TENG to varying frequencies of compression, with a press amplitude of 3 mm. Electrical output, including short-circuit current (ISC) and open-circuit voltage (VOC), was measured using a multifunction meter (UNI-T, UT8805E) and an oscilloscope (OWON, SDS1022). The assembled GT-TENG is depicted in Figure 2.

The operation of the GT-TENG is detailed in Figure 3. Initially, both the friction layer and the electrode are in a state of charge equilibrium. When external force is applied, the friction layers come into contact. Given the higher electronegativity of PTFE compared to TPU, the triboelectrification effect results in the generation of positive charges on the TPU surface and negative charges on the PTFE surface, as illustrated in Figure 3a. Upon removal of the external force, the layers separate, and the connected electrode layer, through electrostatic induction, acquires a charge with opposite electrical properties, leading to a potential difference between the electrodes. This potential difference drives charge transfer through the external circuit, creating a positive current as shown in Figure 3b. When the friction layers are fully separated, the potential difference is at its maximum, as depicted in Figure 3c. Subsequent re-contact under external force reduces the potential difference, reversing the charge flow and generating a negative current, as shown in Figure 3d. This cyclic process of contact and separation induces an alternating current in the external circuit.

To elucidate the working principle of the GT-TENG more effectively, we employed COMSOL Multiphysics software (COMSOL Multiphysics 6.1) to simulate its operation. The results are illustrated in Figure 3e–h, where the lines represent electric field lines and the colors indicate potential distribution. As depicted in Figure 3e, when the friction layers are fully in contact, there is no potential difference between the electrodes, indicating electrostatic equilibrium. Upon removal of the external force and subsequent separation of the friction layers, the potential difference between the electrodes increases as the distance between the layers grows. This increase continues until the layers return to their initial state, at which point the potential difference reaches its maximum. Figure 3f–h show that the potential difference decreases as the friction layers move closer together. These simulations (Figure 3e–h) align with the operational states of the GT-TENG as demonstrated in Figure 3a–d. Figure 3i displays the charge transferred per cycle for the GT-TENG.

## 3. Results

### 3.1. GT-TENG Electrical Output

The output current and voltage signals of the GT-TENG operating at 5 Hz are presented in Figure 4a,b. As the graphene content in the TPU matrix increases, the short-circuit current initially rises, reaching a peak at 0.268 µA with 1.5% graphene, before subsequently declining. Correspondingly, the open-circuit voltage signal exhibits a similar pattern, achieving a maximum value of 110.25 V, also at 1.5% graphene addition. This trend underscores the optimal graphene concentration for enhancing the electrical output of the GT-TENG.

Figure 4c illustrates the amount of charge transferred between electrodes during a single excitation cycle of the GT-TENG. The transferred charge increases with the addition of graphene content in the film, peaking at 0.45 nC when the graphene addition reaches 1.5%. To assess the power output of the GT-TENG, different impedances were introduced into the external circuit. Figure 4d depicts how the power density varies with changes in impedance. As the impedance connected to the external circuit increases, the current markedly decreases while the voltage gradually rises. The power density curves, derived from the measured current and voltage data, indicate that the maximum power density output of the GT-TENG is 54.62 mW/m^2^ at an external resistance of 72 MΩ.

The TPU composite film containing 1.5% graphene and the PTFE were selected as the friction layer materials for the GT-TENG. The voltage signals generated by the GT-TENG under operating frequencies ranging from 1 to 5 Hz, along with capacitor charging curves, are depicted in Figure 5. 

As illustrated in Figure 5a, the GT-TENG was used to charge a 4.7 µF capacitor at various frequencies. At a frequency of 1 Hz, it takes 165 s for the GT-TENG to charge the capacitor to 3 V. As the frequency increases, the charging time decreases; at 5 Hz, the capacitor reaches 3 V in just 67 s, marking a 2.4-fold improvement in charging efficiency compared to 1 Hz. Figure 5b presents the corresponding charging curves, where different capacitors are charged by GT-TENG at 5 Hz. In Figure 5c the voltage output of the GT-TENG increases progressively with frequency, rising from 30 V at 1 Hz to 110.25 V at 5 Hz—a voltage enhancement of approximately 3.5 times. This indicates that the GT-TENG’s electrical output varies significantly with different operating frequencies. These results confirm that the GT-TENG produces distinct electrical signals at different frequencies.

### 3.2. Durability of GT-TENG

The stability and durability of the GT-TENG are crucial considerations in its design. We utilized a TPU composite film containing 1.5% graphene and PTFE as the friction layer materials. To evaluate the GT-TENG’s durability, we conducted long-term operation tests, measuring the strength of current and voltage output signals over an extended period. Additionally, we employed an Atomic Force Microscope (AFM-Nano surf Flex C3000, sourced from Nanosurf Technology Company, Liestal, Switzerland) to examine the surface morphology of the friction layer materials before and after testing. The results of these experiments are depicted in Figure 6.

Figure 6a illustrates that the designed GT-TENG consistently outputs alternating current (AC) with an instantaneous maximum current of approximately 0.268 µA during 11,760 s of continuous operation at a frequency of 5 Hz. Subsequent measurements of the voltage output, as shown in Figure 6b, confirm that the output voltage remains stable and regular even after extended operation, with a maximum value of about 110.25 V, mirroring the initial conditions of the experiment. These results demonstrate that both the current and voltage outputs of the GT-TENG maintain consistent performance over prolonged periods. The GT-TENG exhibits excellent stability and reliability in its electrical output, as evidenced by the continuous and regular signal characteristics after extensive usage.

Figure 6c,e present the initial surface morphology of the TPU and PTFE friction layer materials as measured by an AFM, with initial surface roughness values of 209.4 nm for TPU and 212.6 nm for PTFE. Figure 6d,f depict the surface morphology of these materials after 11,760 s of continuous operation. The surface roughness of the TPU film increased to 133.2 nm, and that of the PTFE film increased to 138.8 nm. Despite this prolonged operation, the changes in surface morphology are minimal, indicating excellent working durability. The GT-TENG continues to produce stable output current and voltage signals, affirming its robust performance over extended use.

### 3.3. Practical Application of GT-TENG

#### 3.3.1. Micro–Nano Power

The friction layers of the GT-TENG consist of a TPU composite film with a graphene content of 1.5% and PTFE, operating at a frequency of 5 Hz. A rectifier bridge is integrated into the external circuit of the GT-TENG to convert the AC generated into direct current (DC). Additionally, a 10 µF capacitor is connected to the circuit to store the electrical energy produced by the GT-TENG. This stored energy is subsequently utilized to power electronic devices. The configuration of the GT-TENG operating circuit system is depicted in Figure 7a.

Figure 7b,c illustrate the setup where the batteries in the sports timer and electronic watch were removed and the GT-TENG was directly connected to the devices. The lower left corners of Figure 7b,c depict how the connections were made after the batteries were removed. Operating at a frequency of 5 Hz, the GT-TENG charged the capacitors to 3 V within 150 s. Upon activating the circuit switch, the GT-TENG provided power to both the sports timer and the electronic watch. The sports timer functioned continuously for 8 s, while the electronic watch operated for 6 s. Additionally, a type F3 green light-emitting diode (LED) was integrated into the GT-TENG circuit. Once the switch was activated and the capacitor reached 3 V, the LED lit up instantly with high luminous intensity, lasting approximately 1 s, as shown in Figure 7d. The capacitor’s voltage immediately dropped to 1.91 V upon discharging and stabilized at this value during GT-TENG operation, as demonstrated in the charge and discharge curves in Figure 7e. When the capacitor voltage stabilized at 1.91 V, the GT-TENG continued to power the LED. Although the luminous intensity of the LED was weaker than initially observed in Figure 7f, it maintained a stable light output.

#### 3.3.2. Conversion of Kinetic to Electrical Energy from Human and Self-Powered Monitoring

The GT-TENG is engineered to harness kinetic energy from human movement and convert it into electrical energy. It is specifically designed for integration into sports shoes and clothing. As the human body moves, the GT-TENG captures kinetic energy and converts it into electrical energy, which powers LEDs to indicate activity. Figure 8a illustrates the installation of the GT-TENG on clothing. When the friction layers of the GT-TENG are separated, the LEDs remain unlit, as depicted in Figure 8b. However, as the body moves, movements such as arm swings or walking cause the friction layer materials within the GT-TENG to contact and then separate. This action converts the motion energy into electrical energy, lighting up the LEDs during the process of contact and separation, as shown in Figure 8c. Similarly, Figure 8d–f illustrate the installation of the GT-TENG on sneakers.

The GT-TENG efficiently harnesses kinetic energy from human movement, converting it into electrical energy, and is capable of generating varied electrical signals based on the amount of kinetic energy captured. This capability allows the GT-TENG to function as a self-powered sensor for monitoring human activities. The operational schematic of the GT-TENG as a self-sustained energy sensor is depicted in Figure 9a. Installed on clothing or sports shoes, the GT-TENG produces distinct electrical signals in response to different human movements. These signals are transmitted by a signal generator connecting the GT-TENG to, for example, a computer or smartphone via Bluetooth, facilitating real-time monitoring of human movements. In the entire Bluetooth wireless transmission system, the signal generator requires battery power, while the others do not require additional power supply. This setup enables continuous tracking and analysis of motion patterns on digital devices.

Figure 9b–d depict the electrical signals generated by the GT-TENG installed in sports shoes during the human movement modes of walking, jogging, and sprinting. These signals are transmitted via Bluetooth to a smartphone, where different electrical signal curves are displayed within the application software. The signal curves presented in Figure 9e–g correspond to the motion modes of walk, jog, and sprint. In the walking state, the electrical signal shown in Figure 9e has a low amplitude and frequency. During jogging, as shown in Figure 9f, both the amplitude and frequency of the electrical signal increase. In the sprinting state, depicted in Figure 9g, the amplitude and frequency of the electrical signal significantly increase. This setup allows for self-powered sensing, enabling the monitoring of human body movement through the signals generated by the GT-TENG.

## 4. Discussion

In this study, we developed a self-driven sensor utilizing a TPU matrix doped with graphene particles, termed GT-TENG. Optimal performance was achieved when graphene was incorporated at a mass fraction of 1.5%. To elucidate the working principles of GT-TENG, we employed COMSOL Multiphysics for simulations of its operational state. Durability tests were conducted on the GT-TENG, and an AFM was utilized to analyze the surface morphology of its friction layer material. The GT-TENG demonstrated the capability to temporarily power sports timers and electronic watches. It can continuously supply energy to an LED and sustain its operation by converting kinetic energy from human movement into electrical energy, thereby enabling autonomous energy supply for monitoring human motion.

The findings of this study align with previous research, such as the work by Zhao Xue [32], who prepared graphene-doped polyimide (PI) composites for TENGs. Zhao found that integrating graphene particles enhanced the electrical output of TENGs, leading to the development of self-powered pressure sensors. Similarly, L. Shooshtari [33] capitalized on the capability of TENGs as self-powered sensors to develop a self-powered SnS_2_/PVK photodetector utilizing a GO/Kapton TENG. Additionally, T. Chomjun [34] created a composite material from natural rubber and activated carbon for TENG applications, which could instantaneously power multiple LEDs and detect human motion. Table 1 presents some TENG information from previous studies. Compared to these studies, our TENG not only continuously supplies energy to an LED but also exhibits superior innovation in motion monitoring and wireless signal transmission through its thoughtful design. Zu [28] fabricated a TENG with excellent durability, which remained stable after 6000 cycles of tests. In this work, the maximum voltage value of GT-TENG changes from 110 V to approximately 108 V with an almost negligible reduction after more than 10,000 cycles.

In this study, we observed that the GT-TENG exhibits optimal electrical output performance when the graphene particle content is 1.5%. It is hypothesized that the layered structure of graphene has a large comparative area, and the graphene dispersed in the TPU matrix act as charge trapping sites, increasing the interface for charge storage, as shown in Figure 1h. Inductive charges are stored in these nanosized graphene particles through the friction process, which increases the surface charge of graphene-doped TPU nanocomposite films, thereby boosting the electrical output of the GT-TENG. However, an excessive addition of graphene nanoparticles can lead to increased dielectric leakage in the composite film, ultimately diminishing the GT-TENG’s electrical output performance.

This study acknowledges certain limitations in the power output of the GT-TENG. While it is capable of continuously supplying energy to an LED, it only powers electronic devices such as sports timers and electronic watches for a brief duration of 6–8 s. Therefore, enhancing the output power of the GT-TENG is essential. Additionally, although we have developed a self-powered sensor for monitoring human motion based on the GT-TENG, it currently only provides a rough estimation of movement patterns. The precision of motion monitoring needs further improvement. In future research, we aim to augment the output power of the GT-TENG through the development of new materials and the surface modification of existing materials. We also plan to enhance the accuracy of our self-powered sensor in monitoring aspects such as pressure and movement frequency.

## 5. Conclusions

We developed a self-powered sensor using a TPU-doped graphene composite film, termed GT-TENG, which harnesses the kinetic energy of human movement to generate its own electrical energy. Experimental testing of its electrical signal output and durability yielded the following conclusions: The GT-TENG exhibits optimal electrical performance when the graphene particle doping in the TPU matrix is at a mass fraction of 1.5%. Under these conditions, the GT-TENG can generate an output voltage of up to 110.25 V, a peak-to-peak current of 0.268 μA, and a maximum power density of 54.62 mW/m^2^ at an external resistance of 72 MΩ, with an effective working area of 20 mm × 20 mm. The sensor demonstrates robust working durability and is capable of temporarily powering small electronic devices and continuously lighting one LED. Furthermore, this self-powered GT-TENG sensor effectively monitors human movements and facilitates wireless signal transmission.

## Figures and Tables

**Figure 1 nanomaterials-14-01549-f001:**
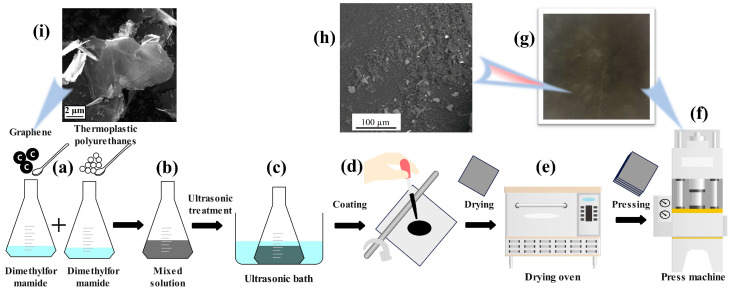
TPU composite film preparation process. (**a**) Putting the graphene particles and TPU particles into DMF solutions. (**b**) Mixing the graphene-doped DMF and TPU-doped DMF solutions. (**c**) Ultrasonic shaking of the mixed solution for uniform distribution of graphene nanoparticles. (**d**) Artificial monolayer film preparation. (**e**) Drying the monolayer film. (**f**) Hot pressing the multilayer film. (**g**) Optical photograph of the multilayer film after hot press. (**h**) Scanning electron micrograph of graphene-doped TPU nanocomposite film. (**i**) Scanning electron micrograph of graphene nanoparticles.

**Figure 2 nanomaterials-14-01549-f002:**
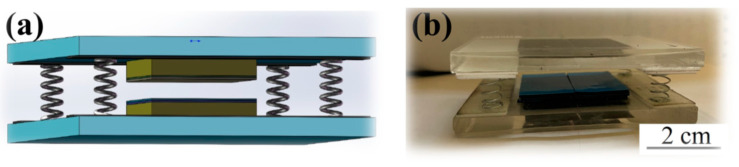
(**a**) Schematic of the GT-TENG model. (**b**) Optical photograph of GT-TENG.

**Figure 3 nanomaterials-14-01549-f003:**
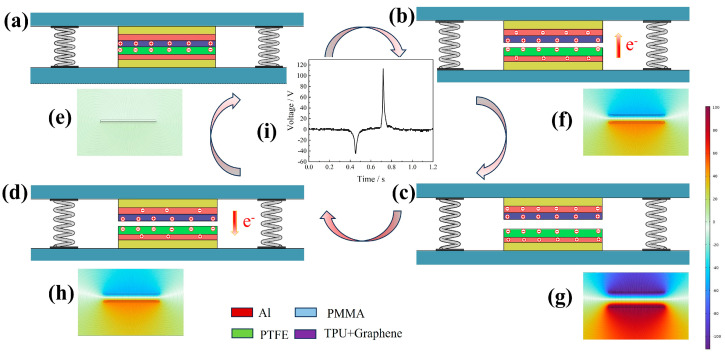
GT-TENG working mechanism diagram. (**a**) Friction layers completely contacted. (**b**) Friction layer material separation process. (**c**) Friction layer completely separated. (**d**) Friction layer material close again. (**e**–**h**) COMSOL simulation GT-TENG working state, which corresponds to figures (**a**–**d**). (**i**) A charge transferred cycle of the GT-TENG.

**Figure 4 nanomaterials-14-01549-f004:**
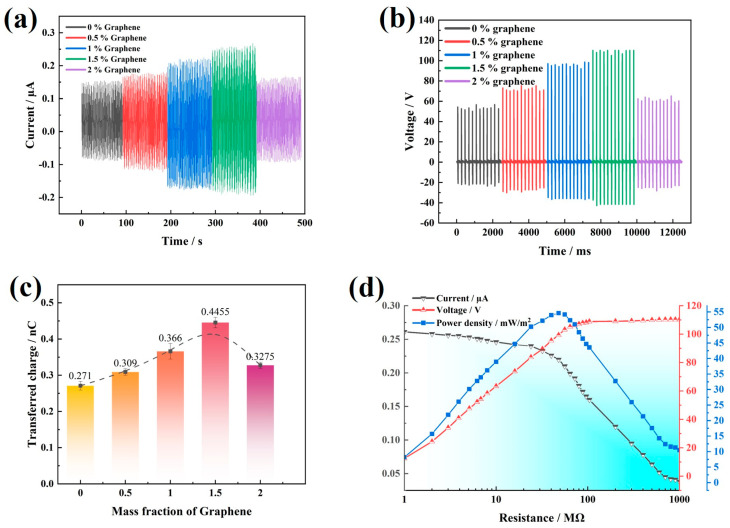
(**a**) Short-circuit current Isc. (**b**) Open-circuit voltage Voc. (**c**) Amount of charge transferred between electrodes in a single generation cycle of GT-TENG. (**d**) GT-TENG output power density regarding resistance.

**Figure 5 nanomaterials-14-01549-f005:**
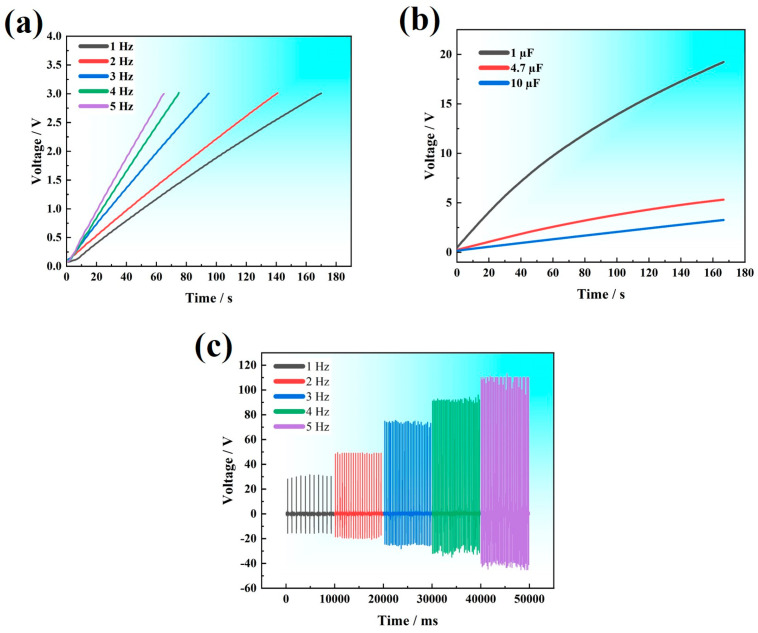
(**a**) Capacitor charging curve at different frequencies. (**b**) Capacitor charging curve with different capacitor. (**c**) GT-TENG voltage signal at different frequency conditions.

**Figure 6 nanomaterials-14-01549-f006:**
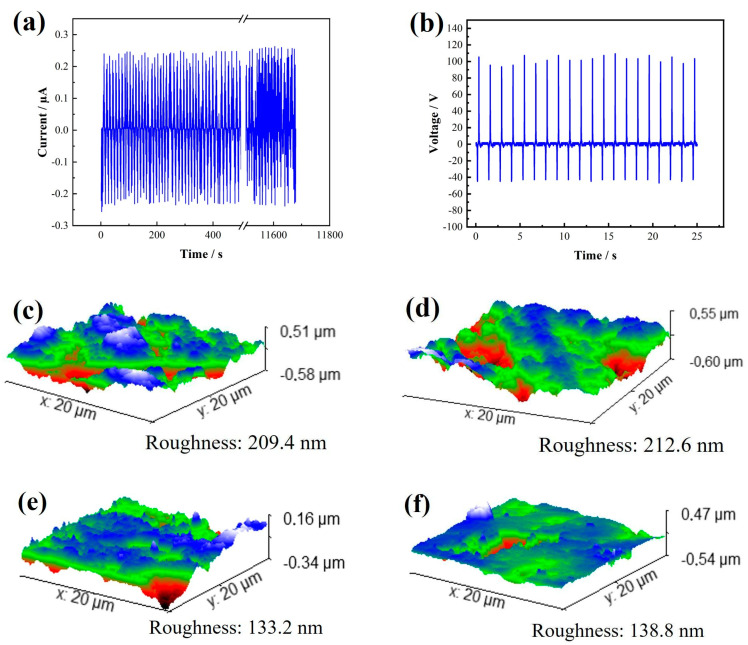
GT-TENG durability test results. (**a**) GT-TENG long-term test Ioc. (**b**) Voc after GT-TENG long-term test. (**c**) AFM photo of initial morphology of TPU composite film. (**d**) AFM photo of surface morphology of TPU composite film after long-term test. (**e**) AFM photo of initial morphology of PTFE film. (**f**) AFM photo of surface morphology of PTFE film after long-term test.

**Figure 7 nanomaterials-14-01549-f007:**
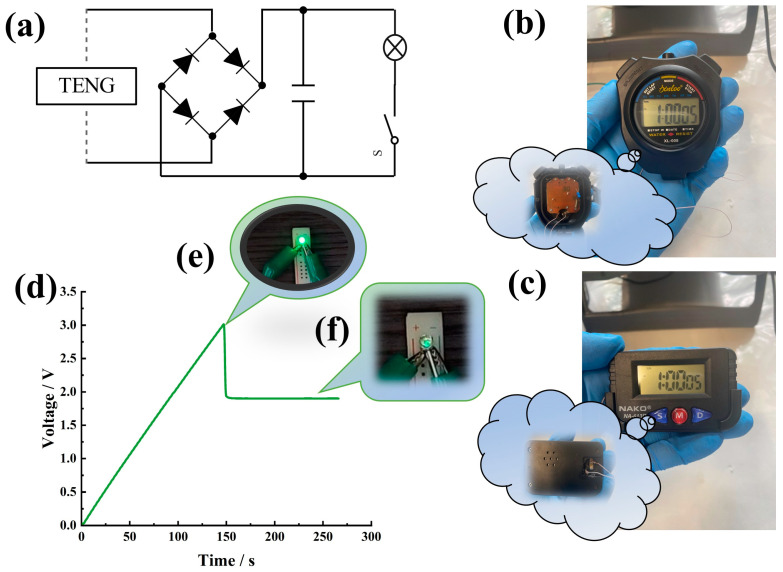
GT-TENG powering small electronic devices. (**a**) Circuit diagram of GT-TENG operation. (**b**) GT-TENG supplying energy for sports timer. (**c**) GT-TENG supplying energy for electronic watch. (**d**) Voltage profile of GT-TENG lighting one LED. (**e**) LED instantaneous illumination. (**f**) LED continuous illumination.

**Figure 8 nanomaterials-14-01549-f008:**
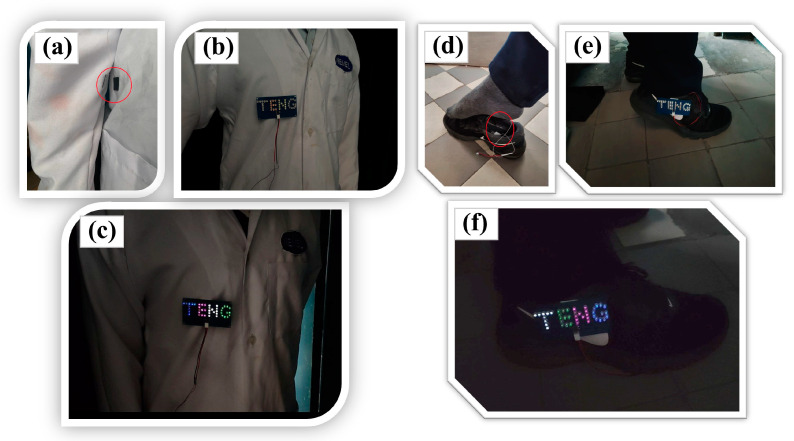
GT-TENG collects human kinetic energy. (**a**) The installation of the GT-TENG on clothing. (**b**) The LEDs remain unlit without movement. (**c**) The LEDs lighted with movement. (**d**) The installation of the GT-TENG on sneaker. (**e**) The LEDs remain unlit without movement. (**f**) The LEDs lighted with movement.

**Figure 9 nanomaterials-14-01549-f009:**
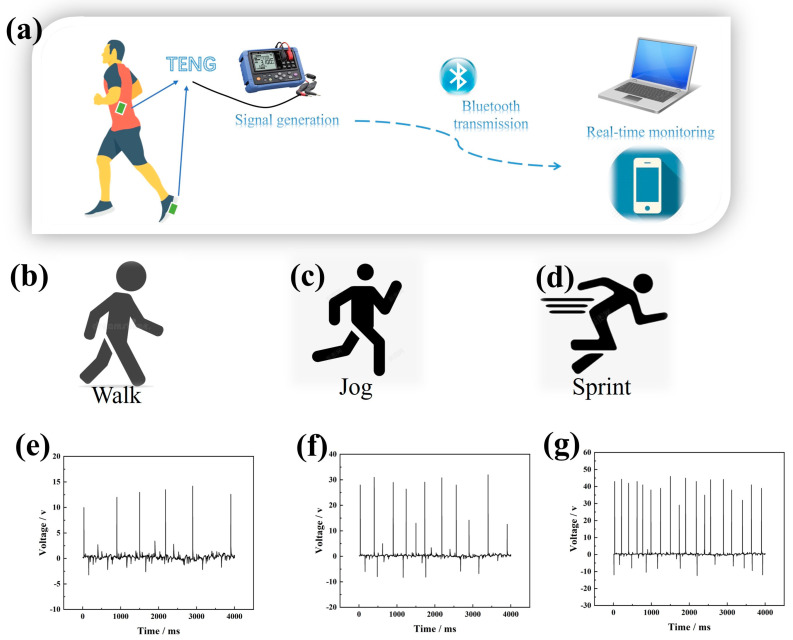
GT-TENG monitoring human body kinetic energy. (**a**) Schematic diagram of GT-TENG wireless monitoring operation. (**b**) Sport mode—walk. (**c**) Sport mode—jog. (**d**) Sport mode—sprint. (**e**) Signal generated by GT-TENG in sport mode of walk. (**f**) Signal generated by GT-TENG in sport mode of jog. (**g**) Signal generated by GT-TENG in sport mode of sprint.

**Table 1 nanomaterials-14-01549-t001:** TENGs information from previous studies.

Matrix Material	Doping Material	Maximum Power/Power Density	Durability	Application	Reference
Polyimide	Graphene	0.22 mW	Not tested	Self-Powered Pressure Sensor	[32]
Kapton	Graphene Oxide	10 mW/cm^2^	Not tested	Self-Powered Photodetector	[33]
Rubber	Activated carbon	242 mW/m^2^	Not tested	Motion-Sensing	[34]
Silicon Rubber	MoS2/GO	1.3 mW	Stable	Power Wearable Electronics	[28]
Polyurethane	Graphene	54.62 mW/m^2^	Excellent	Wireless Self-Powered Sport Sensor	Our Work

## Data Availability

Data will be made available on request.

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
