# Peer review of "Graphene-Doped Thermoplastic Polyurethane Nanocomposite Film-Based Triboelectric Nanogenerator for Self-Powered Sport Sensor"

_nanomaterials, 2024, doi:10.3390/nano14191549_

Round 1

Reviewer 1 Report

Comments and Suggestions for Authors

The manuscript describes a triboelectric nanogenerator (TENG) which uses a graphene-doped thermoplastic polyurethane film and PTFE as friction materials and has been applied for sport sensor.

In the abstract the authors claim an exceptional working durability. 

This can be a strong point of this work.

However, although the experiments in Fig. 6 are certainly relevant, the discussion on state of the art and comparison with previous works are not satisfactory.

Please, add an exhaustive discussion in the introduction on previous work that addressed this key point and also a comparison between quantitative figures of merit of previous works and of this work. 

From the title it would seem that the main contributions of this work is both at material level and at the applicative level (sport sensor). Is this correct?

Or perhaps the novelty is at material level and sport sensor is just an application used as proof-of-concept? (In this case I would suggest to remove from title to keep it more general).

The introduction should contain an in-depth literature survey of previous works on TENGs where doping of the friction layers has been used (of course it is correct to talk about the most relevant previous works in the discussion, but a complete literature survey in the introduction is mandatory).

The authors write: "Many researchers have noted that the intricate designs and expensive materials involved in TENG construction elevate production costs, hindering mass production and limiting the practicality of TENGs."

To me this sentence is questionable as many ultra-low cost TENGs have been reported. The authors may want to revise or add more details and strong references to support this statement.

Please, define all acronyms (e.g. GT-TENG) when using the first time in the Introduction also.

Please, increase the font size for all text, labels,... in figures 1, 4, 5, 6a, 6b, 7d. In general, make sure all text is readable without magnification.

In figure 4, please, also show at least some representative measurements (raw values, as measured) of the open circuit voltage and short circuit current as a function of time (to show the peak,...). This should be before figure 4 (which is an elaboration on the raw data), so that the results which are currently shown in Fig. 4 may be more clearly discussed (e.g. peak values...).

For figure 4, dfferent from text, the caption show mass fractions above 1 which is clearly a typo. Please, revise. 

The text should discuss the sub-figures in the order of the sub-figures (e.g. 8a, 8b, 8c, 8d,...) and not in a different order.

Please, carefully revise Fig. 9 to improve clarity.

Is the complete wearable system (including bluetooth etc) totally battery-less?

If so, please, provide a diagram.

Sub-figure (c) is not clear, please, provide clear images and raw data (e.g. with supplementary information).

Typos or revisions

Please, use a consistent style for figure captions, sub-figures,... (and make sure it is suitable for the journal). (e.g. Fig. 1 uses i., ii., I, II, Fig 2, 4, 6 use (a), (b), Fig. 3 same as Fig 1,..., Fig. 7 and Fig. 9 a mix of (a) (b) and i ii... 

Please, use the same style for all figures (e.g.  Fig. 6 the font size used for (a), (b) is different from (c-f).

Comments on the Quality of English Language

please, see comments to authors

Author Response

Comments and Suggestions for Authors:

Comment 1: The manuscript describes a triboelectric nanogenerator (TENG) which uses a graphene-doped thermoplastic polyurethane film and PTFE as friction materials and has been applied for sport sensor. In the abstract the authors claim an exceptional working durability. This can be a strong point of this work. However, although the experiments in Fig. 6 are certainly relevant, the discussion on state of the art and comparison with previous works are not satisfactory. Please, add an exhaustive discussion in the introduction on previous work that addressed this key point and also a comparison between quantitative figures of merit of previous works and of this work.

Responce 1: Thank you for recognizing our work. Following your suggestion, we have added a discussion on the durability of friction nanogenerator operation in previous work in the Introduction section, see lines 65-77 of the paper. We have also drawn a table in line 369 of the paper to compare this paper with previous work to fully demonstrate the advantages and innovations of this work.

Comment 2: From the title it would seem that the main contributions of this work is both at material level and at the applicative level (sport sensor). Is this correct? Or perhaps the novelty is at material level and sport sensor is just an application used as proof-of-concept? (In this case I would suggest to remove from title to keep it more general).

Responce 2: Thank you for your critical thinking. This work contributes both at material level and at the applicative level. The results of the experiments have shown that the designed TENG can work as self-powered sensor to detect human movement, not just stay at the level of proof-of-concept.

Comment 3: The introduction should contain an in-depth literature survey of previous works on TENGs where doping of the friction layers has been used (of course it is correct to talk about the most relevant previous works in the discussion, but a complete literature survey in the introduction is mandatory).

Responce 3: Thank you for pointing out this. We agree with this comment. Therefore, we have added the literature survey of nanomaterial doping mothed in TENG design, please see line 65-72.

Comment 4: The authors write: "Many researchers have noted that the intricate designs and expensive materials involved in TENG construction elevate production costs, hindering mass production and limiting the practicality of TENGs." To me this sentence is questionable as many ultra-low cost TENGs have been reported. The authors may want to revise or add more details and strong references to support this statement.

Responce 4: We apologize that this sentence didn't make sense, and we've removed the sentence according to your comment.

Comment 5: Please, define all acronyms (e.g. GT-TENG) when using the first time in the Introduction also.

Responce 5: Thank you for pointing out this. We have defined all the acronyms according to your comments.

Comment 6: Please, increase the font size for all text, labels, in figures 1, 4, 5, 6a, 6b, 7d. In general, make sure all text is readable without magnification.

Responce 6: We apologize for the inconvenience of reading these pictures, We have improved all the images based on your suggestions.

Comment 7: In figure 4, please, also show at least some representative measurements (raw values, as measured) of the open circuit voltage and short circuit current as a function of time (to show the peak,...). This should be before figure 4 (which is an elaboration on the raw data), so that the results which are currently shown in Fig. 4 may be more clearly discussed (e.g. peak values...).

Responce 7: Thank you for pointing out this. We agree with this comment. We have added the raw date of open circuit voltage and short circuit current in Figure 4 (a)(b), please see line 212.

Comment 8: For figure 4, different from text, the caption shows mass fractions above 1 which is clearly a typo. Please, revise.

Responce 8: Thank you for pointing out this. We have revised this by your suggestion.

Comment 9: The text should discuss the sub-figures in the order of the sub-figures (e.g. 8a, 8b, 8c, 8d,...) and not in a different order.

Responce 9: Thank you for pointing out this. We have revised this by your suggestion. Please see line 304-314.

Comment 10: Please, carefully revise Fig. 9 to improve clarity.

Responce 10: Thank you for pointing this out. We agree with this comment. As the wireless signal receiving software used in the design of the sensing and detection system is yet to be perfected, the received electrical signals present irregular curves with insufficient quantization, and it is necessary to start from the software to improve the signal reception and presentation. In the next work, we will improve the monitoring system design on the basis of this TENG, systematically study the accuracy of this TENG as a self-supplied energy sensor to monitor the frequency, pressure and other parameters of the movement, and fully quantify the monitoring accuracy.

Comment 11: Is the complete wearable system (including Bluetooth etc) totally battery-less?

If so, please, provide a diagram.

Responce 11: Throughout the wearable system, the Bluetooth signal generator connected to the TENG requires a miniature battery to power it.

Comment 12: Sub-figure (c) is not clear, please, provide clear images and raw data (e.g. with supplementary information).

Responce 12: We mounted the GT-TENG on clothes, the electricity generated when friction occurs can light up the LED, but the LED brightness is relatively small. In order to show that the LED glows under friction, we need to take pictures in a darker space. We have tried this several times and the image will always be a little unclear.

【Typos or revisions】

  1. Please, use a consistent style for figure captions, sub-figures,... (and make sure it is suitable for the journal). (e.g. Fig. 1 uses i., ii., I, II, Fig 2, 4, 6 use (a), (b), Fig. 3 same as Fig 1,..., Fig. 7 and Fig. 9 a mix of (a) (b) and i ii...

Thank you for pointing out this. We agree with this comment. We have revised this by your suggestion.

  1. Please, use the same style for all figures (e.g. Fig. 6 the font size used for (a), (b) is different from (c-f).

Thank you for pointing out this. We agree with this comment. We have tried to revise this by your comment, but the images obtained from the AFM have their source format and exact consistency is not available for the time being.

Reviewer 2 Report

Comments and Suggestions for Authors

In this study, the authors developed a triboelectric nanogenerator (TENG) based on graphene and TPU composites, with PTFE as the substrate. The device demonstrated excellent power generation capabilities and continuous stability, and the authors successfully verified its potential as a power source for small devices in wearable applications, indicating strong potential for real-world use. The data presented in the article are robust, the discussion is comprehensive, and the overall quality aligns with the standards of Nanomaterials. However, a few minor issues require further clarification and improvement.

  1. The authors mentioned AI applications in the abstract and introduction; however, the article does not adequately reflect scenarios involving AI. In practice, it is challenging to directly combine TENG with AI applications due to its limited power output, which is insufficient to meet the demands of even the smallest AI chips. TENGs are better suited as simple signal-testing sensors or as components within larger wearable IoT or AI systems, where they handle small device power supply and signal acquisition. The relevant statements in the article should be revised for greater accuracy.

  2. In Figure 3, the contrast between different materials is minimal, affecting readability. The authors should consider using higher contrast colors to improve clarity.

  3. There are some concerns regarding the strength of the TPU and graphene composite material. For the long-term test in Figure 6, it would be beneficial if the authors could calculate the force applied in each test and compare it to the force or stress generated when the human body steps on it. This would help ensure that the test conditions more accurately reflect real-world application scenarios.

Author Response

Comments and Suggestions for Authors:

In this study, the authors developed a triboelectric nanogenerator (TENG) based on graphene and TPU composites, with PTFE as the substrate. The device demonstrated excellent power generation capabilities and continuous stability, and the authors successfully verified its potential as a power source for small devices in wearable applications, indicating strong potential for real-world use. The data presented in the article are robust, the discussion is comprehensive, and the overall quality aligns with the standards of Nanomaterials. However, a few minor issues require further clarification and improvement.

Comment 1: The authors mentioned AI applications in the abstract and introduction; however, the article does not adequately reflect scenarios involving AI. In practice, it is challenging to directly combine TENG with AI applications due to its limited power output, which is insufficient to meet the demands of even the smallest AI chips. TENGs are better suited as simple signal-testing sensors or as components within larger wearable IoT or AI systems, where they handle small device power supply and signal acquisition. The relevant statements in the article should be revised for greater accuracy.

Response 1: Please forgive us for the lack of clarity in the text and thank you for your valuable comments, we have revised the statement about AI in the text as you requested, see line 26-33 in the text.

Comment 2: In Figure 3, the contrast between different materials is minimal, affecting readability. The authors should consider using higher contrast colors to improve clarity.

Response 2: Thank you for pointing this out. We agree with this comment. Therefore, we have redrawn the picture, and higher contrast colors have been chosen to improve clarity. Please see Figure 3 in line 188.

Comment 3: There are some concerns regarding the strength of the TPU and graphene composite material. For the long-term test in Figure 6, it would be beneficial if the authors could calculate the force applied in each test and compare it to the force or stress generated when the human body steps on it. This would help ensure that the test conditions more accurately reflect real-world application scenarios.

Response 3: Thank you for pointing this out. Your proposal is very valuable to us and is part of our work program. The focus of this work is to design a triboelectric nanogenerator based on a composite thin film and then experimentally demonstrate that this TENG has the ability to act as a sport sensor and has a very good working durability. In our next work, we plan to investigate in detail the sensing accuracy of the TENG-based sensors, including parameters such as frequency of motion monitoring and, as you suggested, pressure monitoring.

Reviewer 3 Report

Comments and Suggestions for Authors

The manuscript of Yang et al. describes interesting results about the development of TENGs based on graphene-doped thermoplastic polyurethane nanocomposite. However, to be acceptable for publication, there are points to be improved, as follows:

Please revise the language.

The introduction must be revised to be more concise in repeated sentences.

 Please improve the quality of the images in Fig. 1 and its description in the Figure caption.

Please correct the Figure caption of Fig .3: “COMSOL simulation GT-TENG working state, and 185 the figure - V corresponds to.”

Figure 9 is confusing. Please quantify the results shown in the inset and improve the discussion.

A Table for comparison is necessary to highlight the novelty and advantages of incorporating the TENG.

Please correct the order of Figures: “Figure 5. (a) Capacitor charging curve, (b) GT-TENG voltage signal at different frequency conditions.” Also, it would be interesting to provide results for different capacitors under the same excitation frequency.

In addition to the AFM images in Fig. 6, the authors could provide the KPFM profile for sample.

Please provide the charge transferred per cycle for the TENG.

Comments on the Quality of English Language

Please revise the English language. There are sentences in the Introduction that must be shortened and revised.

Author Response

Comments and Suggestions for Authors:

The manuscript of Yang et al. describes interesting results about the development of TENGs based on graphene-doped thermoplastic polyurethane nanocomposite. However, to be acceptable for publication, there are points to be improved, as follows:

  1. Please revise the language.

Thank you for your suggestion. We have invited a native English speaker to carefully review and revise the manuscript in terms of grammar, style, and syntax. I believe this will greatly improve the quality of the manuscript.

  1. The introduction must be revised to be more concise in repeated sentences.

Thank you for pointing this out. We agree with this comment. Therefore,we have revised the preamble to shorten sentences and reduce repetitive sentences to make the presentation more concise

  1. Please improve the quality of the images in Fig. 1 and its description in the Figure caption.

We apologize for the unclear image and inaccurate description of the steps, we have modified the image and improved the description of the experiment steps according to your suggestions. Please see line 137-144.

  1. Please correct the Figure caption of Fig .3: “COMSOL simulation GT-TENG working state, and 185 the figure Ⅱ - V corresponds to.”

Please forgive the error caused by my carelessness, we have fixed the error as you requested. Please see line 190-192.

  1. Figure 9 is confusing. Please quantify the results shown in the inset and improve the discussion.

Thank you for pointing this out. We agree with this comment. As the wireless signal receiving software used in the design of the sensing and detection system is yet to be perfected, the received electrical signals present irregular curves with insufficient quantization, and it is necessary to start from the software to improve the signal reception and presentation. In the next work, we will improve the monitoring system design on the basis of this TENG, systematically study the accuracy of this TENG as a self-supplied energy sensor to monitor the frequency, pressure and other parameters of the movement, and fully quantify the monitoring accuracy.

  1. A Table for comparison is necessary to highlight the novelty and advantages of incorporating the TENG.

Thank you for your kind suggestion. We have added a table in line 370 to highlight the novelty and advantages of the fabricated TENG by your suggestion.

  1. Please correct the order of Figures: “Figure 5. (a) Capacitor charging curve, (b) GT-TENG voltage signal at different frequency conditions.” Also, it would be interesting to provide results for different capacitors under the same excitation frequency.

Please forgive the error caused by my carelessness, we have fixed the error as you requested. Please see line 234-234. And also, the results for different capacitors under the same excitation frequency is provided in line 234 according to your suggestion.

  1. In addition to the AFM images in Fig. 6, the authors could provide the KPFM profile for sample.

Your suggestion is very useful, but due to the limitation of laboratory conditions, KPFM test can not be carried out for the time being, in the next research work, we will add the analysis method according to your suggestion.

  1. Please provide the charge transferred per cycle for the TENG.

Thank you for your kind suggestion. We have added the charge transferred cycle in Figure 3

Comments on the Quality of English Language

  1. Please revise the English language. There are sentences in the Introduction that must be shortened and revised.

Thank you for pointing this out. We have revised the preamble to shorten sentences and reduce repetitive sentences to make the presentation more concise

Round 2

Reviewer 1 Report

Comments and Suggestions for Authors

The manuscript has been partially improved, but some comments have not been properly addressed.

The previous "comment 1" was

"In the abstract the authors claim an exceptional working durability. This can be a strong point of this work. However, although the experiments in Fig. 6 are certainly relevant, the discussion on state of the art and comparison with previous works are not satisfactory. Please, add an exhaustive discussion in the introduction on previous work that addressed this key point and also a comparison between quantitative figures of merit of previous works and of this work."

The authors claim in the abstract an "exceptional working durability".

If it is "exceptional", the statement has to be solidly supported.

The working durability can be quantified by proper figures of merit (e.g. xyz % reduction of the open circuit peak voltage after xyz cycles), but a quantitative comparison with literature is completely missing.

There is no discussion of literature on durability, but the "exceptional" claim requires in-depth exhaustive discussion with quantitative comparison.

The previous "comment 2" was

"The introduction should contain an in-depth literature survey of previous works on TENGs where doping of the friction layers has been used (of course it is correct to talk about the most relevant previous works in the discussion, but a complete literature survey in the introduction is mandatory)."

However the discussion of literature seems to miss some relevant works where doping of the friction layers has been used.

The previous "comment 11" was

"Is the complete wearable system (including Bluetooth etc) totally battery-less?"

The authors now clarify that "the Bluetooth signal generator connected to the TENG requires a miniature battery to power it." This should be clearly discussed in the manuscript.

A previous comment was 

"Please, carefully revise Fig. 9 to improve clarity.

... Sub-figure (c) is not clear, please, provide clear images and raw data (e.g. with supplementary information)."

Apparently the authors recognize that the quality of the figure is poor but may not improve it. 

However, these are standard measurements (routinely shown in so many previous works) and the quality of presentation must be sufficient for publication.

Comments on the Quality of English Language

Please, improve english and correct all typos (e.g. "powerty" (Table 1), revise table 1 (it is better to avoid hyphen in tables, write each word in a single line),...

Author Response

The manuscript has been partially improved, but some comments have not been properly addressed.

Comment 1: The previous "comment 1" was

 "In the abstract the authors claim an exceptional working durability. This can be a strong point of this work. However, although the experiments in Fig. 6 are certainly relevant, the discussion on state of the art and comparison with previous works are not satisfactory. Please, add an exhaustive discussion in the introduction on previous work that addressed this key point and also a comparison between quantitative figures of merit of previous works and of this work."

The authors claim in the abstract an "exceptional working durability".

If it is "exceptional", the statement has to be solidly supported.

The working durability can be quantified by proper figures of merit (e.g. xyz % reduction of the open circuit peak voltage after xyz cycles), but a quantitative comparison with literature is completely missing.

There is no discussion of literature on durability, but the "exceptional" claim requires in-depth exhaustive discussion with quantitative comparison.

Response 1: Please excuse any inaccuracies in our wording. We intended to convey that the TENG was designed with excellent durability. However, during the English touch-up process, the term "exceptional" was used by the native English-speaking editor. We have since corrected this. A discussion of the TENG's durability has been added to the text; please refer to line 19-20, 80-89, 379-385 for details.

Comment 2: The previous "comment 2" was

 "The introduction should contain an in-depth literature survey of previous works on TENGs where doping of the friction layers has been used (of course it is correct to talk about the most relevant previous works in the discussion, but a complete literature survey in the introduction is mandatory)."

However the discussion of literature seems to miss some relevant works where doping of the friction layers has been used.

Response 2: Please forgive us for not refining this very well. The text has now been refined according to your suggestions, please refer to line 66-79.

Comment 3: The previous "comment 11" was

 "Is the complete wearable system (including Bluetooth etc) totally battery-less?"

The authors now clarify that "the Bluetooth signal generator connected to the TENG requires a miniature battery to power it." This should be clearly discussed in the manuscript.

Response 3: Thank you for pointing this out, we have changed the text as you suggested, please refer to line 339-341.

Comment 4: A previous comment was

 "Please, carefully revise Fig. 9 to improve clarity.

 ... Sub-figure (c) is not clear, please, provide clear images and raw data (e.g. with supplementary information)."

Apparently the authors recognize that the quality of the figure is poor but may not improve it.

However, these are standard measurements (routinely shown in so many previous works) and the quality of presentation must be sufficient for publication.

Response 4: Thank you for pointing this out, we have changed the text as you suggested, added the raw data and tried to make the picture as clear as possible, please refer to line 354-359.

Comment 5: Comments on the Quality of English Language

Please, improve English and correct all typos (e.g. "powerty" (Table 1), revise table 1 (it is better to avoid hyphen in tables, write each word in a single line),

Response 5: Thank you for pointing out these errors, we have corrected the text as you suggested, please refer to line 386.

Reviewer 3 Report

Comments and Suggestions for Authors

In view of the response provided by the authors, I consider that manuscript can be accepted as is.

Author Response

Comment: In view of the response provided by the authors, I consider that manuscript can be accepted as is.

Response: Thank you very much for recognizing our work!